# New Sensor Device to Accurately Measure Cable Tension in Cable-Driven Parallel Robots

**DOI:** 10.3390/s21113604

**Published:** 2021-05-21

**Authors:** Guillermo Rubio-Gómez, Sergio Juárez-Pérez, Antonio Gonzalez-Rodríguez, David Rodríguez-Rosa, Lis Corral-Gómez, Alfonso I. López-Díaz, Ismael Payo, Fernando J. Castillo-García

**Affiliations:** School of Industrial and Aerospace Engineering, University of Castilla-La Mancha, 45071 Toledo, Spain; guillermo.rubiogomez@uclm.es (G.R.-G.); sergio.juarez@uclm.es (S.J.-P.); antonio.gonzalez@uclm.es (A.G.-R.); david.rrosa@uclm.es (D.R.-R.); lis.corral@uclm.es (L.C.-G.); Alfonso.Lopez@uclm.es (A.I.L.-D.); Ismael.Payo@uclm.es (I.P.)

**Keywords:** tension measurement, cable-driven parallel robots, strain gauges

## Abstract

Cable-driven parallel robots are a special type of robot in which an end-effector is attached to a fixed frame by means of several cables. The position and orientation of the end-effector can be controlled by controlling the length of the cables. These robots present a wide range of advantages, and the control algorithms required have greater complexity than those in traditional serial robots. Measuring the cable tension is an important task in this type of robot as many control algorithms rely on this information. There are several well-known approaches to measure cable tension in cable robots, where a trade-off between complexity and accuracy is observed. This work presents a new device based on strain gauges to measure cable tension specially designed to be applied in cable-driven parallel robots. This device can be easily mounted on the cable near the fixed frame, allowing the cable length and orientation to change freely, while the measure is taken before the cable passes through the guiding pulleys for improved accuracy. The results obtained from the device show a strong repeatability and linearity of the measures

## 1. Introduction

Measuring stresses with strain gauges is very common in engineering. This technique has undoubted advantages but also has some disadvantages, specifically when applied to determine the tension in cables. The aim of this work is to propose a novel device to measure the tension in cables that are suitable for application in cable-driven parallel robots (CDPRs).

CDPRs are a special type of parallel manipulator where rigid links are replaced by cables; therefore, the end-effector is sustained by a set of *n* cables [1]. The length of each cable is controlled by means of a motor-winch set, usually located on a fixed frame. By controlling the cable lengths, the position (e.g., Khosravi and Taghirad [2]) and/or orientation (e.g., Tadokoro et al. [3]) of the end-effector can be controlled.

Some of the advantages that these manipulators provide compared to both serial manipulators and conventional parallel ones are the following:CDPRs present the most light-weight structure for a manipulator from the point of view of structural design [1].The motors only have to move the payload and the cables; therefore, CDPRs are able to move a much higher mass or to employ less energy.CDPRs can cover large workspaces, since very long cables are easy to wind [4].

Applications of cable force measurements on CDPRs are extensive. One of the most important drawbacks of CDPRs is the complexity of the robot dynamics and its strong nonlinear behavior [5]. This is due to the cable tension, which varies nonlinearly with the end-effector position and orientation. Due to this dynamic feature, the control effort required to achieve a high level of precision in positioning/orientation of the robot is very high [6,7]. To achieve this, the most commonly employed control strategies are based on linearization of the nonlinear dynamic terms, usually by means of feedforward linearization [8,9,10,11] or by using inverse dynamics techniques [12].

In order to effectively employ the aforementioned approaches, the end-effector position must be precisely measured. Additionally, this measure must be carried out with a high enough time resolution to be used by the control loop, which should be within the order of milliseconds. To solve this, the most common procedure is to use the motor’s encoder readings to estimate the end-effector position through kinematic models [13]. This allows for a high speed reading but, since the end-effector position is not directly measured but is just estimated, it yields position errors that cannot be neglected for most practical applications.

Another approach is to directly measure the end-effector position using computer vision techniques [14]. This method is accurate but has difficulty obtaining the required time resolution for the measures as a correct resolution requires processing a high amount of data.

In constrained CDPR, i.e., with the same number of cables as degrees of freedom of the end-effector [15], the cable tension distribution allows us to compute the end-effector position as long as tension measures are highly accurate. High speed measures can be obtained by using well-known force measurement principles.

Additionally, in overconstrained CDPR, i.e., with more cables than degrees of freedom of the end-effector [16], there are an infinite number of possible cable force distributions for a single end-effector pose; this means that force control algorithms, which rely on cable force measurements, are commonly used for adjusting tension levels to a feasible value [17,18]. Moreover, cable force measurements are required in CDPR for implementing contact control [19,20] or load identification [21] algorithms.

In summary, an accurately measure of cable tension in CDPR is required for estimating end-effector pose or/and for dynamics control purposes.

Some commercial off-the-shelf devices that could be located at the frame to measure cable tension can be found in the industry (see Figure 1) but their performance in range, resolution, or sensory latency make them unsuitable for application to CDPRs.

In that sense, most of the CDPR prototypes that include cable tension sensors present customized solutions. A relevant work is presented by Kraus et al. [22]. This paper compares the three main ways for estimating cable tension by means of force sensors (see Figure 2).

Some examples of estimating cable tension by means of a force sensor located at the pulley of the frame or end-effector, directly in the cable (see Figure 2a), are [22,23,24]. Winch-integrated sensors (see Figure 2b) can be found in [25,26,27]. Finally, we found that only in Scalera et al. [28] is the three-pulley concept (see Figure 2c) implemented for measuring the cable tension of CDPR.

The first type of sensor can be placed directly on the end-effector (see Figure 2a), providing a more accurate measure as the force being applied to the end-effector is directly measured; however, it requires a wire cable supply or non-wire signal (with its correspondent large latency and lack of synchronism). This type of sensor can also be placed on the guiding pulleys of the actuators, as in [20,29,30]. As the sensor is attached to the robot frame, it must be placed in a position where the cable orientation does not change when the robots moves. Additionally, another drawback of this approach is that the sensor is placed after one or some guiding pulleys; therefore, the measured tension differs from the real tension due to the friction of the pulleys.

Winch-integrated sensors (see Figure 2b) simplify the integration into the mechanical design but provide some errors in the force such that the end-effector suffers owing to the friction losses of the winch and all pulleys.

Finally, the three-pulley concept (see Figure 2c) can also be placed at the point of interest but the inertia of the actuator is slightly increased by the inertia of each pulley that rotates during robot maneuvers. The friction of the pulleys can also be a disadvantage yielding non-accurate measurements.

All of the commercial-off-the-shelf sensors (Figure 1) or the ones based on the three-pulley concept (e.g., Scalera et al. [28]) are based on measuring the compression efforts of the link that supports the central pulley. Figure 3 illustrates the conceptual idea of both commercial and customized devices based on the three-pulley concept.

This work proposes a new sensor device that employs strain gauges to measure cable tension based on the three-pulley concept (see Figure 2c). Conceptualization of the proposal is based on estimating cable tension, not by measuring compression efforts but flexion ones by means of strain gauges (see Figure 4).

For the same cable tension, flexion efforts are significantly higher than compression ones, and therefore, for the same sensor system, the amplitude of the acquired signal is bigger, the resolution of the sensor is higher, and the typical noise of force cells or strain gauges has less influence on the acquired signal.

This device is specifically designed to be easily employed in CDPR, as it allows the cable to run freely through the device while it can be fixed near the robot frame and before any guiding pulley, avoiding errors caused by pulley friction.

## 2. Materials and Methods

### 2.1. Mechanical Approach

To accurately measure the cable tension, we propose to measure the deformation caused by the cable in the bars that support a set of three pulleys. These three pulleys redirect the cable from its original rectilinear path. In this way, the higher the cable tension, the higher the deformation of these bars.

Figure 5 shows the schematic of the device. The cable is labeled 1, the bar where the strain gauges are located is labeled 2, the piece that supports the redirecting pulley is labeled 3, while the pulleys are labeled 4. Parts 2 and 3 are connected with a pair of bolts in points *P* and *Q*. Cable tensions are marked with arrows and the letter *T*. Three pulleys are used to guide the cable, all of them with radius *r*.

The forces produced in bar 2 by these cable tensions, along with its correspondent bending moment and axial force diagrams, are shown in Figure 6a).

Note that the bending moment M=aT and the axial force N=T are constant in the area between points *P* and *Q*. In this area, the maximum normal stress occurs on the upper side of the section and the minimum occurs on the lower face. These values are as follows:(1)$$\begin{array}{cc} \sigma_{xx}^{max} & =\;\left(\frac{1}{he}+\frac{6a}{eh^{2}}\right)T\\ \sigma_{xx}^{min} & =\;\left(\frac{1}{he}-\frac{6a}{eh^{2}}\right)T \end{array}$$
where *e* is the thickness of bar 2 and where *a* and *h* are the geometrical characteristic of bar 2, shown in Figure 6b. Equation (Equation 1) shows that the stress is proportional to *T*. The stress $\sigma_{xx}=E\;\epsilon_{xx}$ is proportional to the deformation $\epsilon_{xx}$ through Young’s modulus *E* since $\sigma_{yy},\sigma_{zz}=0$.

The dimensions of bar 2 must fulfill the hypotheses of the Euler Bernoulli beam in the Theory of Strength of Materials [31], which results in the stresses in (Equation 1) being only approximate. For this reason, an experimental calibration of the device is necessary. However, the tension between points *P* and *Q* is approximately constant, making this an optimal area to place the gauges.

The last important issue of the mechanics is the minimum allowed radius of the pulleys, which affects miniaturization of the device. The minimum radius, $r_{min}$, avoids the plasticity of the cable. Cable bending in the pulley gives the flexion stress in the cable. To obtain reasonable life from the device, a proper diameter for the pulley must be chosen. In general, the larger the size of the pulley with respect to the wire diameter, the longer the service life. Manufacturers provide tables with the minimum recommended pulley diameter. Let us denote $d_{c}$ as the diameter of the cable. Without further information, a minimum ratio, $r_{min}\geq 30\cdot d_{c}$ can be assumed to guarantee a reasonable cable life cycle.

### 2.2. Electronics and Instrumentation

As shown in (Equation 1), both $\sigma_{xx}^{max}$ and $\sigma_{xx}^{min}$ are proportional to the cable tension through a constant $k=\frac{1}{he}\pm \frac{6a}{eh^{2}}$ that has two terms: $k_{1}=\frac{1}{he}$, which represents the stress due to axial tension *N*, and $k_{2}=\frac{6a}{eh^{2}}$, which represent the stress caused by bending moment *M*.

The stress caused by bending moment *M* is considerably higher than that caused by axial tension as long as a>h, since $\frac{k_{2}}{k_{1}}=\frac{6a}{h}$. Therefore, the strain gauges are placed to measure *M* and to cancel *N*; this is achieved by positioning the strain gauges on the device as shown in Figure 7.

Figure 8 shows the signal conditioning electronics, where *V* is the bridge voltage, *R* is a fixed value for resistances, $e_{0}$ is the bridge output, $Rg_{1}$ is the upper strain gauge, and $Rg_{2}$ is the lower strain gauge.

The conditioning circuit yields the following expression for the bridge output:(2)$$e_{0}=\frac{V}{2}K_{s}\cdot \left(\epsilon_{1}-\epsilon_{2}\right)$$
where $K_{s}$ is the gauge factor and $\epsilon_{i}$ is the deformation of the gauge $Rg_{i}$. Since $\epsilon_{1}=\frac{\sigma_{xx}^{max}}{E}$ and $\epsilon_{2}=\frac{\sigma_{xx}^{min}}{E}$, Equation (Equation 2) can be rewritten as follows:(3)$$e_{0}=\frac{V}{2E}K_{s}\cdot \left(2\frac{6a}{eh^{2}}T\right)$$

The signal conditioning electronics employed therefore allow us to cancel both of the axial strain effects and to multiply the bending moment effects by a factor of 2, reducing possible noise in the signal.

The bridge voltage, *V*, can be therefore acquired by a digital data acquisition device. If the resolution of the DAQ device is *n* bits, the measurement resolution, i.e., the minimum change that can be measured, is as follows:(4)$$\Delta V_{m}=\frac{V_{max}-V_{min}}{2^{n}}$$
with $V_{max}$ and $V_{min}$ being the maximum and minimum allowed measurements of the analog input of the DAQ device, respectively.

### 2.3. Prototype

A prototype of the device was built in order to analyze the repeatability, sensibility, and linearity of the measurement system as well as to obtain the calibration curve. The final device is shown in Figure 9. As can be seen, bars 2 and 3 in Figure 5 are duplicated on both sides in order to make a symmetrical system.

The prototype dimensions in Figure 6b) are a=38 mm, b=44 mm, h=12 mm, d=5 mm, and e=6 mm. This yields a relation between $K_{2}$ and $K_{1}$ of $\frac{k_{1}}{k_{2}}=19$, ensuring that stress generated by the bending moment is significantly higher than that generated by axial tension. The gauges employed have a gauge factor of $K_{s}=2$. The signal provided by the electronic system is registered with a NI USB-6341 (National Instruments, Austin, TX, USA) data acquisition board in which the analog input channels have a resolution of 16 bits. Since $e_{0}$ lies in the range 0–5 V, the board range was set to ±5 V, yielding a voltage reading resolution of $0.153\times 10^{-3}$ V.

## 3. Validation and Results

### 3.1. Experiments Protocol for Validation

The experimental setup shown in Figure 10 was employed for validation and calibration of the tension measurement prototype.

To apply a known load to the device, weight plates of 10, 15, and 20 kg were hanged from a 20 cm steel cable of 2.5 mm diameter. Every weight plate and the additional elements employed to attach them was weighted with a GRAM DSX-30 (GRAM DX, l’Hospitalet de Llobregat, Barcelona, Spain) precision weighing scale with a resolution of 2 g. Table 1 shows the masses employed for the validation experiments.

To assess the repeatability of the measurement system, for each mass, five consecutive measures were taken, each one after releasing the mass, reloading, and waiting long enough for the device to reach a stationary state. Each measure was obtained by acquiring 1000 samples during 1 s and by calculating the mean voltage value. To ensure that the mean was a representative value of the sample data, the one-sample Kolmogorov–Smirnov test [32] was applied to one randomly selected measure out of each mass test. For every test, the null hypothesis suggesting that the data come from a standard normal distribution is not rejected at the 5% significance level. Figure 11 shows a comparison of the cumulative distribution function (CFD) of the sample data to that of a standard normal distribution.

### 3.2. Repeatability

The repeatability of the measurement system is assessed by means of the box plot [33] of the measures made for each mass point. In order to compare the repeatability of measures from different tests, the box plot is obtained from the values normalized to the mean value. The results are shown in Figure 12.

As can be observed, for every test, the maximum difference with the mean value of the values between percentiles 25th and 75th is less than 0.5%. Additionally, it can be observed that there are no outliers in the values of any test and that the median value of each test has a difference with a mean value of less than 0.1%.

### 3.3. Linearity and Calibration

In Figure 13, the measurements are represented versus the corresponding weights along with the first-order polynomial fitted to the experimental data. As can be observed, the linearity is very strong since $R^{2}=0.9984$ and the relation between voltage and mass is as follows:(5)M=20778.1×V+5234.4

### 3.4. Dynamics Analysis

The previous sections demonstrated that the device yields accurate measurements of cable tension under static conditions.

This section preliminarily analyses the influence of natural movement of the robot during its maneuvers on the cable tension measurement. In order to demonstrate that tension remains constant during the cable during travel and that the friction loses can be considered negligible, the following experiment was carried out. A mass of 10 kg was attached to a blocked winch, which was suddenly released, allowing the mass to freely fall under the effect of gravity. A sketch of the experiment setup is shown in Figure 14.

The mass was released from a height of 2.5 m, and measurements were taken until the mass collided with the floor; however, for the sake of readability, only the measurements up to 200 ms before the collision are shown. A set of 10 consecutive experiments was carried out.

To assess the coherence of the obtained experimental results, a simple dynamic model was employed, which considers the winch inertia and viscous friction, the mass, and its friction with air but neglects the friction in the sensor pulleys. The model parameters are shown in Table 2; these parameters were obtained experimentally by means of a GRAM DSX-30 (GRAM DX, l’Hospitalet de Llobregat, Barcelona, Spain) precision weighing scale and an incremental quadrature encoder OMROM E6B2-CWZ6C (OMROM, Osaka, Japan) with a resolution of 1000 ppr.

Figure 15 shows the experimental results together with the simulation results. The *x*-axis was adjusted so that the moment when tension starts changing matches for each test and the simulation. As can be observed, experimental results mainly match the results obtained from the simulation, validating the assumption that sensor pulley friction can be neglected for dynamic measures. The differences observed at the beginning of the tests can be explained by the differences in the way that the winch was released, since this was done manually. Additionally, a very good concordance between the steady state values and the original static value of the tension can be observed, which suggests that the rotation of the sensor pulleys has little influence on the measurement. The oscillations that can be observed from 0.2 s can be explained as the weight inevitably oscillates during the fall. This effect does not occur in a real cable robot.

## 4. Conclusions

In this work, a novel device to measure tension in cables, specially designed for application in cable-driven parallel robots was proposed.

The mechanical principle and the device design were detailed along with the required electronics and instrumentation. In this paper, a prototype with a range from 100 N to 650 N was built, but the design presented here can be easily scaled for other tension ranges.

Finally, the results of repeatability, calibration, and linearity of the device were exposed. Regarding the repeatability of the device, the maximum deviation along all measures with respect to the mean is 0.6%. The measures show high linearity as the $R^{2}$ value of the fitting to a first-order polynomial is 0.9984.

## Figures and Tables

**Figure 1 sensors-21-03604-f001:**
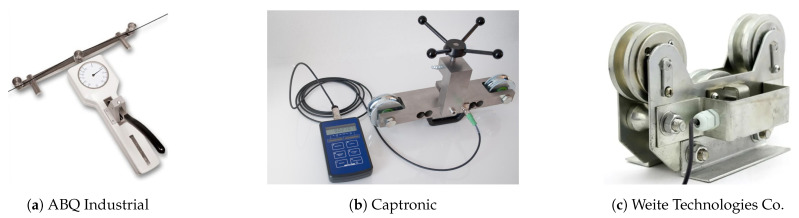
Examples of commercial off-the-shelf devices for measuring cable tension.

**Figure 2 sensors-21-03604-f002:**
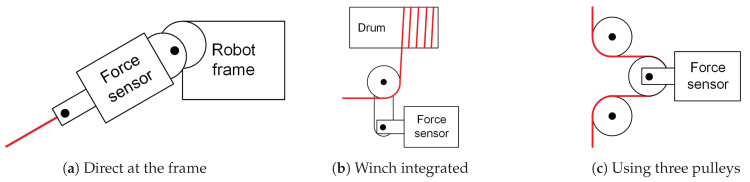
Examples of commercial off-the-shelf devices for measuring cable tension.

**Figure 3 sensors-21-03604-f003:**
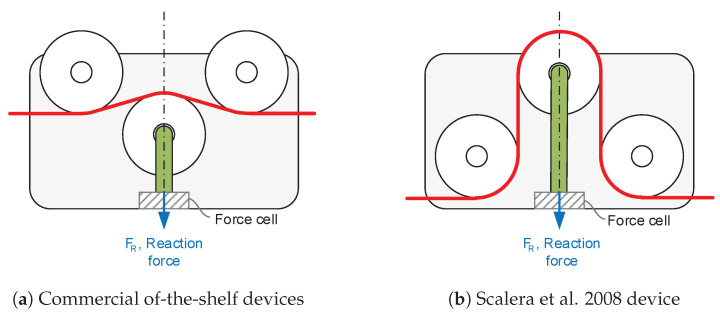
Three-pulley devices based on measuring compression efforts.

**Figure 4 sensors-21-03604-f004:**
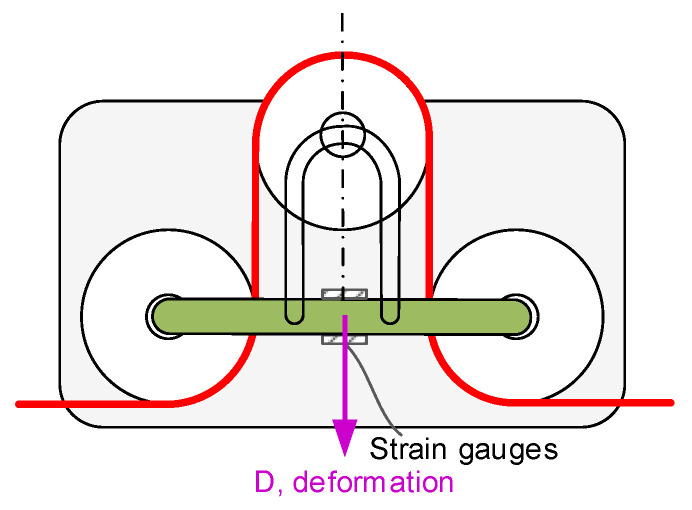
Conceptualization of our proposal.

**Figure 5 sensors-21-03604-f005:**
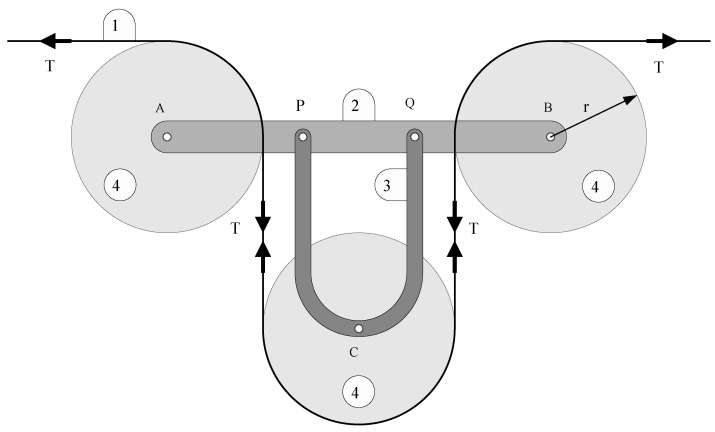
Scheme of the cable tension-measuring device. T = cable tension, r = pulley radius, A,B,C = pulley centers, P,Q = connection points between bars 2 and 3.

**Figure 6 sensors-21-03604-f006:**
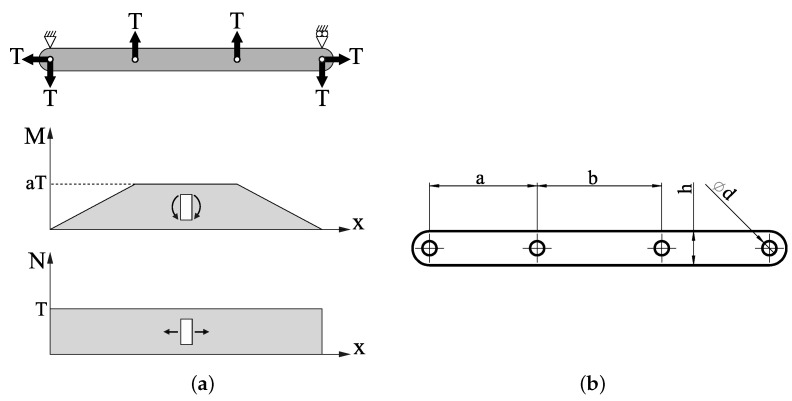
(**a**) Forces on the measuring bar. (**b**) Geometry of the measuring bar.

**Figure 7 sensors-21-03604-f007:**
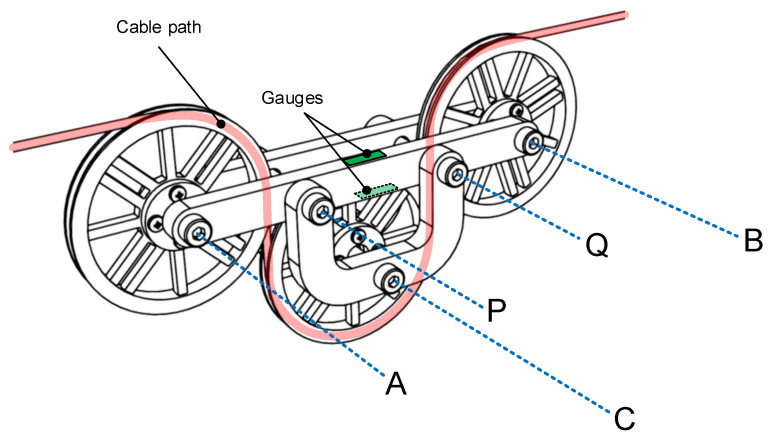
Strain gauges position for bending moment measurement. A,B,C = pulley centers, P,Q = connection points between bars 2 and 3.

**Figure 8 sensors-21-03604-f008:**
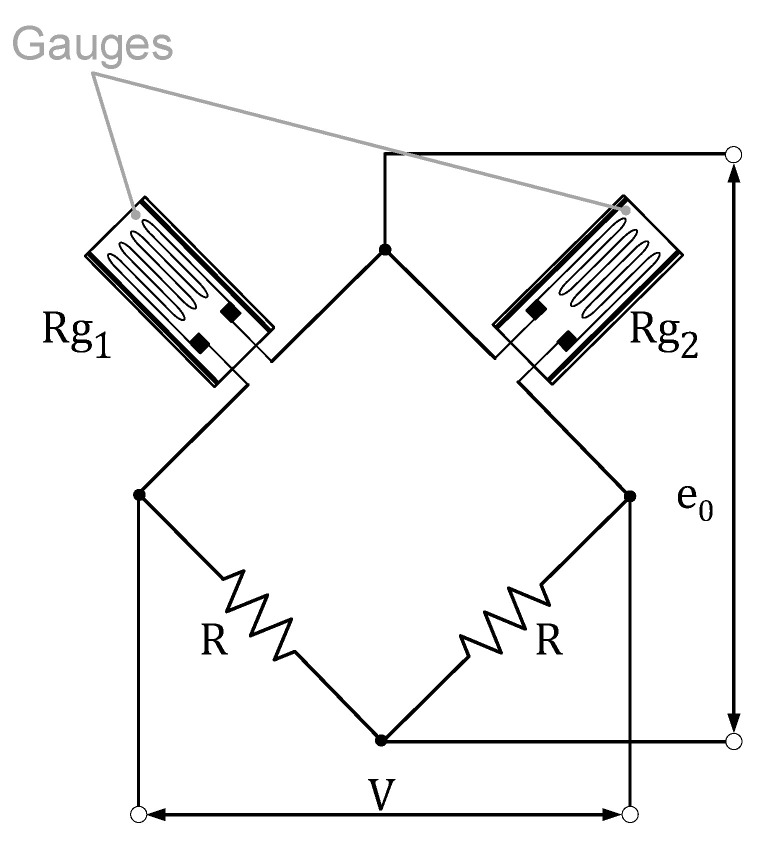
Signal conditioning electronics. R = fixed resistance, $Rg_{1}$ = upper strain gauge, $Rg_{2}$ = lower strain gauge, V = supply voltage, $e_{0}$ = output voltage.

**Figure 9 sensors-21-03604-f009:**
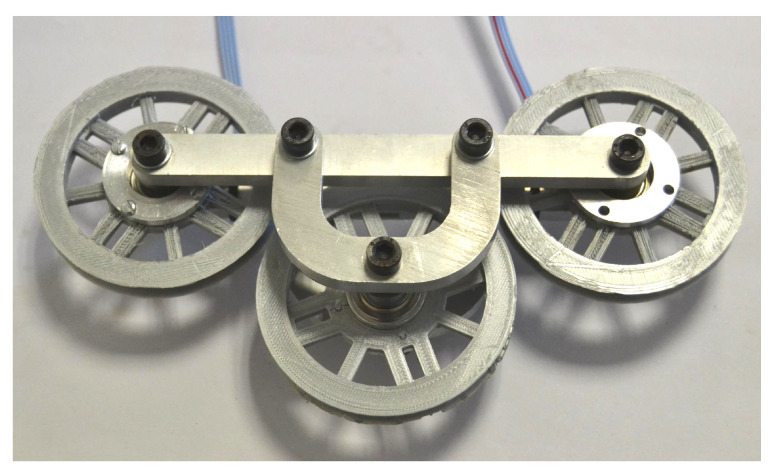
Prototype for validation.

**Figure 10 sensors-21-03604-f010:**
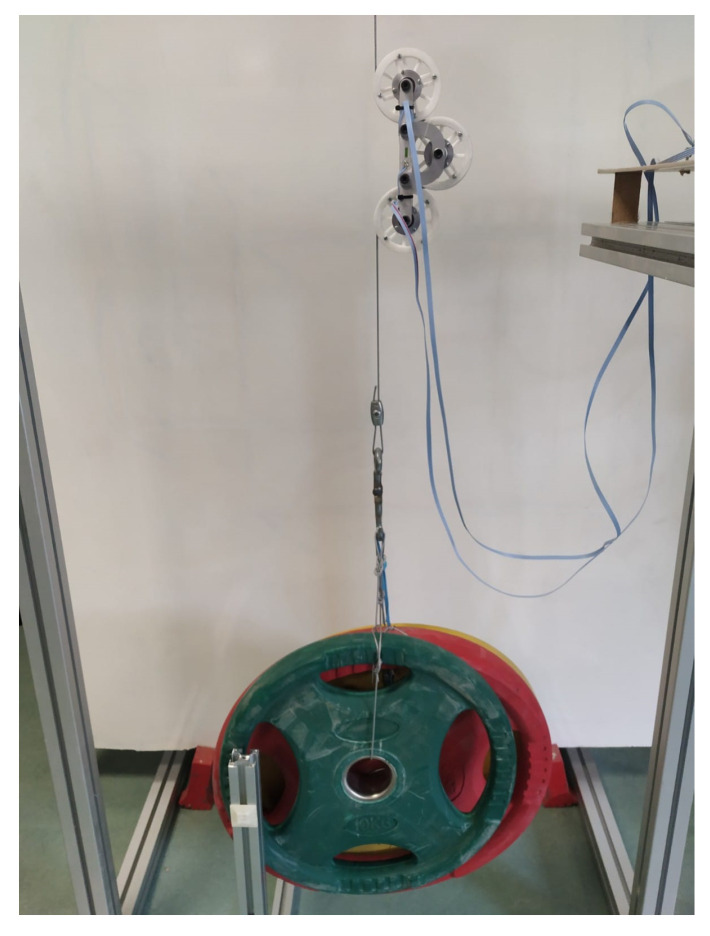
Experimental setup for validation.

**Figure 11 sensors-21-03604-f011:**
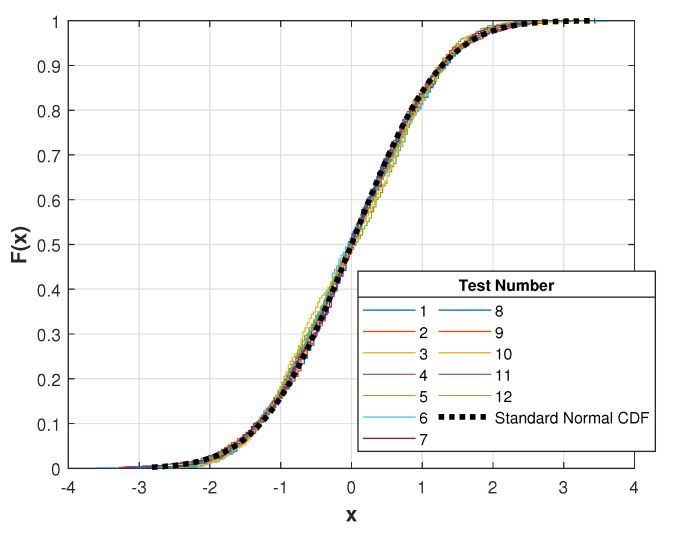
CDF for empirical data.

**Figure 12 sensors-21-03604-f012:**
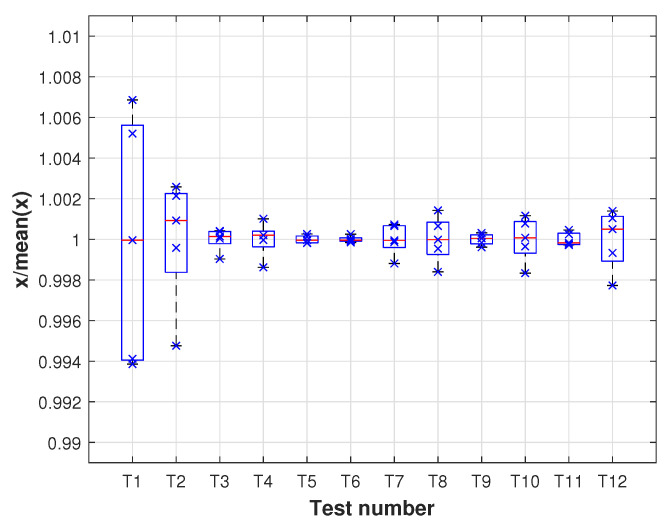
Repeatability results.

**Figure 13 sensors-21-03604-f013:**
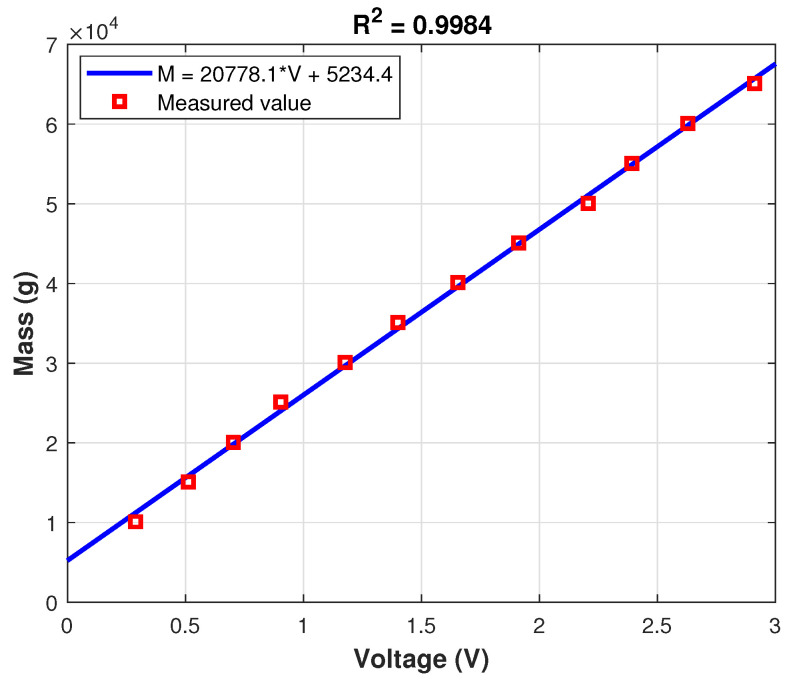
Calibration results.

**Figure 14 sensors-21-03604-f014:**
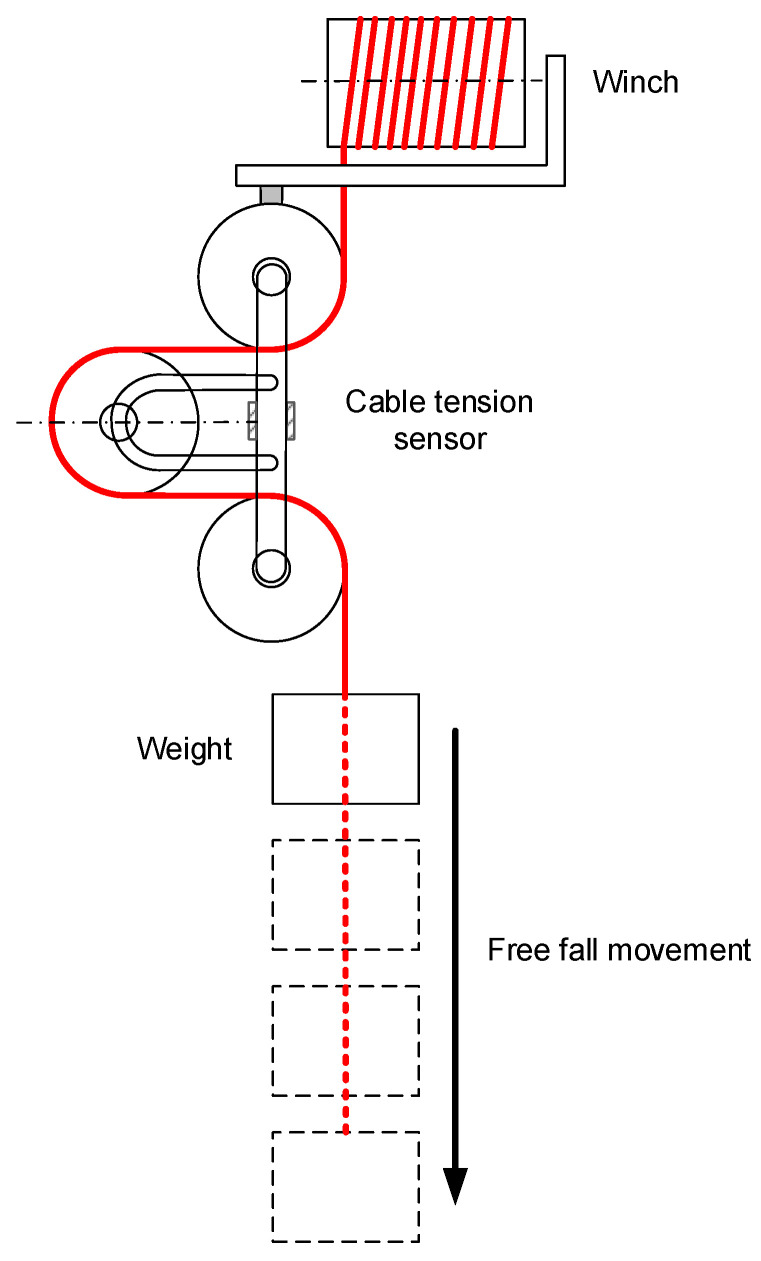
Sketch of the dynamic test approach.

**Figure 15 sensors-21-03604-f015:**
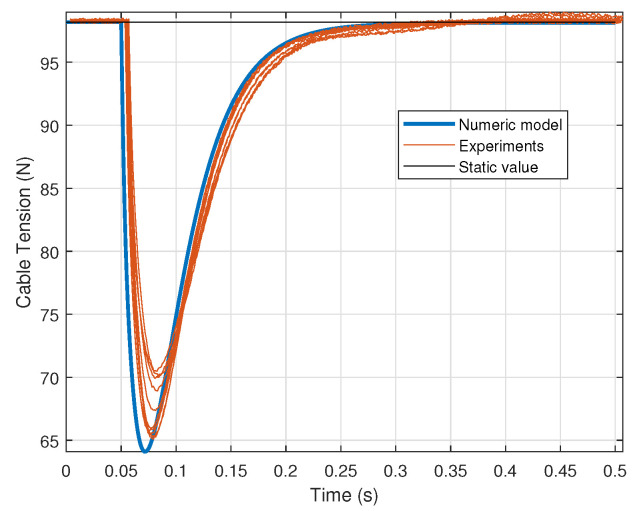
Results of the dynamic test.

**Table 1 sensors-21-03604-t001:** Weights employed for device validation and calibration.

Test Number	1	2	3	4	5	6	7	8	9	10	11	12
**Mass (g)**	10,103	15,105	20,074	25,118	30,088	35,089	40,087	45,102	50,095	55,096	60,094	65,110

**Table 2 sensors-21-03604-t002:** Weights employed for device validation and calibration.

Parameter	Units	Value
Weight mass	Kg	10.103
Weight air friction coefficient	Ns/m	0.0102
Winch mass	Kg	0.0465
Winch radius	m	0.06
Winch Inertia	Kg × m${}^{2}$	0.000837
Winch viscous friction	Nm × s	0.000837

## Data Availability

Data available on request due to restrictions eg privacy or ethical. The data presented in this study are available on request from the corresponding author. The data are not publicly available due to research group policy.

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
