# Peer review of "New Sensor Device to Accurately Measure Cable Tension in Cable-Driven Parallel Robots"

_sensors, 2021, doi:10.3390/s21113604_

Round 1

Reviewer 1 Report

The authors present a variation of a known mechanism for measuring tension in cables. Bibliographical analysis is incomplete in that it fails to mention previous research which presented a similar concept. Please see my following remarks:

- Row 50: infinitely

- Among the cited methods to measure tension I find that the authors fail to mention the measurement of the tension directly on the cable described by Scalera et al. "Cable-Based Robotic Crane (CBRC): Design and Implementation of Overhead Traveling Cranes Based on Variable Radius Drums" (2018) IEEE Transactions on Robotics, 34 (2), pp. 474-485. The system the authors implemented is strikingly similar in concept with what the authors of the IEEE paper propose (see Fig. 11 in the paper), where a load cell measures the deformation of the central pulley support structure. An in-depth discussion on the differences shall be icluded in the paper, with emphasis on the innovations the authors propose.

- Along the same lines as the previous remark. The title should do without the "Novel" term, since this is simple variation of the mechanism presented by Scalera et al. in 2018. This is not to say that the paper is not novel entirely, only that it should be put into perspective considering the cited previous work.

- I feel that in light of the previous remarks, the paper lacks a proper contribution to the state of the art. Some suggestions: a study on the influence of the friction of the pulleys in the measurement of the cable tension; measurement of cable tension with a continuously moving cable (with/without acceleration). Including either of these would add to the state of the art considerably. This last concept in particular is one that is important for cable robots, since with respect to the support structure, the cable moves during actuation, as the authors correctly consider.

The authors did a moderately good job at describing their prototype; however they should better acknowledge previous research and add some considerations on non-trivial measurements as per my last remark. Thus, I consider the paper publishable prior major revisions.

Reviewer 2 Report

The paper presents a new device to measure the cable tension for wired robots.

The paper is well written. The presented device is very simple, which is why I am concerned with regard to the originality of the results.

In conclusion, please specify the main differences between the device presented in your paper and the ones below:

Wire Rope Tension Type Three Pulley Load Cell Sensor for Crane Measuring System (https://wtau123.en.made-in-china.com/product/WwhJtcuHgQVq/China-Wire-Rope-Tension-Type-Three-Pulley-Load-Cell-Sensor-for-Crane-Measuring-System.html)

or

https://www.captronic-sensor.com/mesure-de-force-de-tension,us,8,53.cfm

or

https://www.straightpoint.com/clamp-on-line-tensiometer

or

https://www.abqindustrial.net/store/tension-meters-c-98/tension-meters-mechanical-c-98_42/dnw-tension-meter-for-pretensioned-wire-fiber-ropes-p-1159.html

What is your main contribution, with regard to the already existing cable tension measuring devices, used or not for cable driven parallel robots?

Round 2

Reviewer 1 Report

The authors responded satisfactorily to all my remarks. In my opinion the paper is in a state that is acceptable for publication.

Reviewer 2 Report

The paper has been considerably improved and so it can be published